# Domain Generalization: A Tale of Two ERMs

## Abstract

Domain generalization (DG) is the problem of generalizing from several distributions (or domains), for which labeled training data are available, to a new test domain for which no labeled data is available. A common finding in the DG literature is that it is difficult to outperform empirical risk minimization (ERM) on the pooled training data. In this work, we argue that this finding has primarily been reported for datasets satisfying a *covariate shift* assumption. When the dataset satisfies a *posterior drift* assumption instead, we argue that "domain-informed ERM," wherein feature vectors are augmented with domain-specific information, outperforms pooling ERM. These claims are supported by a theoretical framework and experiments on language and vision tasks.

## 1 Introduction: the ERM dilemma in domain generalization

Domain generalization (DG) is a learning problem where the learner has access to labeled data from several source domains, and the goal is to generalize to a new target domain for which no labeled data is available. Let $X$ denote the input features, $Y$ the label, and $D$ the domain index.

A persistent puzzle in DG is the surprising effectiveness of empirical risk minimization (ERM), a baseline that simply pools labeled data from all source domains together and trains a domain-agnostic classifier. Despite extensive efforts to design sophisticated DG algorithms, multiple studies have consistently shown that ERM remains highly competitive:

- Gulrajani and Lopez-Paz [2021] (empirical): "when carefully implemented and tuned, ERM outperforms the state-of-the-art in terms of average performance... no algorithm included in DomainBed (dataset) outperforms ERM by more than $1\%$."

- Rosenfeld et al. [2021] (theory): "IRM and its alternatives fundamentally do not improve over standard Empirical Risk Minimization."

- Teterwak et al. [2025] (empirical): "the additional tuning in our improved baseline ERM++ outperforms both the prior ERM baselines and all recent SOTA methods on DomainBed."

Similar findings about the strong performance of ERM have been reported across other datasets and settings [Koh et al., 2021, Sagawa et al., 2022, Bai et al., 2024].

A related observation is that most existing DG approaches learn a classifier that predicts $Y$ solely from $X$, thus ignoring domain information during inference. This is reflected in recent surveys:

- Wang et al. [2022] (survey): "The goal of domain generalization is to learn a robust and generalizable predictive function $h : \mathcal{X} \to \mathcal{Y}$ from the $M$ training domains to achieve a minimum prediction error on an unseen test domain $S_{test}$."

- Zhou et al. [2023] (survey): "The goal of DG is to learn a predictive model $f : \mathcal{X} \to \mathcal{Y}$ using only source domain data such that the prediction error on an unseen domain $T = \{x^T\}$ is minimized."

This is despite the fact that early works on DG learn predictions based not only on feature vectors, but also on domain-specific information [Blanchard et al., 2011, Muandet et al., 2013].

In this work, we argue that conclusions about ERM being "hard to beat" stem primarily from the fact that most benchmark DG datasets are from vision tasks. These datasets are characterized by a *covariate shift* assumption, which means that there is a single classifier that performs well on all domains, and only the marginal distribution of $X$ changes from domain to domain. In such applications, strong performance is indeed possible without the use of domain-specific information, and a domain-agnostic classifier can be trained by ERM on the pooled training data.

Furthermore, we study DG problems characterized by *posterior drift*, where the conditional distribution of $Y|X$ (i.e., the *posterior*) changes with domain. We argue that for such DG problems, pooling ERM is inadequate, and stronger performance is achievable by "domain-informed" ERM, where domain specific information is used both during training and at inference.

The contributions of this work are:

- A theoretical framework extending the original formulation of DG by Blanchard et al. [2011].
- Risk bounds that characterize when domain-specific information is beneficial (posterior drift) and when it is not (covariate shift).
- A quantification of the difference between domain generalization and domain adaptation, addressing an open question in Blanchard et al. [2021, Lemma 9].
- Empirical validation of these findings on both language and vision tasks.

## 2 Literature review

Blanchard et al. [2011] introduced the domain generalization (DG) problem, motivated by a medical application involving the automatic gating of flow cytometry data. Since then, most DG research has focused on applications in computer vision. A typical DG task in this setting involves training models on labeled images from multiple visual domains (e.g., styles or rendering conditions) and evaluating generalization to a previously unseen domain. Benchmark datasets such as VLCS [Fang et al., 2013], PACS [Li et al., 2017], OfficeHome [Venkateswara et al., 2017], DomainNet [Peng et al., 2019], and ImageNet-Sketch [Wang et al., 2019] have become standard in this line of work.

In these vision-based setups, the underlying distributional shift can be described as covariate shift [Ben-David et al., 2006, Mansour et al., 2009], where the marginal distribution $P_X$ varies significantly across domains—often with disjoint support. Importantly, domain information is frequently viewed as irrelevant or even spurious [Sagawa et al., 2020] for predicting labels. Consequently, much of the literature has focused on learning domain-invariant representations [Sun and Saenko, 2016, Ganin et al., 2016, Arjovsky et al., 2019]. Additional references are in Appendix A.

In contrast, our work is motivated by a different class of problems characterized by *posterior drift* [Scott, 2019, Cai and Wei, 2021, Maity et al., 2024, Zhu et al., 2024], where the conditional distribution of $Y|X$ varies across domains. This type of shift commonly arises in natural language processing (NLP) tasks. For a given input sentence $X$, different annotators—or populations—may interpret its semantic content differently, leading to divergent labels $Y$ (e.g., offensive vs. non-offensive, positive vs. negative). Such inherent ambiguity in language often results in systematic disagreement in annotations. Empirical studies have documented these effects across a wide range of NLP tasks [De Marneffe et al., 2019, Plank, 2022, Deng et al., 2023].

## 3 Domain generalization: A general probabilistic formulation

In standard classification, a random pair $(X, Y)$ is assumed to be drawn from a fixed joint distribution $P_{XY}$, where $X \in \mathcal{X}$ is a feature vector and $Y \in \mathcal{Y} = \{1, \ldots, K\}$ denotes the corresponding class label[1]. The goal is to learn a function $f : \mathcal{X} \to \mathcal{Y}$ that minimizes the risk:

$$\mathbb{E}_{(X,Y) \sim P_{XY}} \left[ \mathbb{1}_{\{f(X) \neq Y\}} \right].$$

---

[1]This section easily extends to regression, but subsequent sections are focused on classification.

Domain generalization (DG) can be framed in a similar way. Let $\mathcal{D}$ denote a set of possible domains, where the term *domain* is a synonym for a joint distribution of $X$ and $Y$. Let $D$ be a random variable on $\mathcal{D}$. Furthermore, let $M$ be a random variable on a space $\mathcal{M}$ that, intuitively, provides partial information about $D$. The idea in DG is that $D$ determines a distribution of $(X, Y)$, but is not observed. $M$ provides partial information about $D$, and is thus useful at test time in adapting the classifier to the test domain. While the choice of $M$ will depend on the application, one choice that is always viable is to take $M$ to be $P_{X|D}$, the marginal distribution $X$ for the given domain, which is known at test time though the unlabeled test sample. As we argue below, the observability of $M$ is what makes DG an interesting problem, and distinct from standard classification.

Formally, we assume that $(X, Y, M, D)$ are jointly distributed, with joint distribution denoted as $P_{XYMD}$. This distribution induces several other distributions of interest in this paper. We follow convention in denoting marginal distributions by keeping the relevant subscripts. For example, $P_{XYD}$ denotes the joint distribution of $(X, Y, D)$ after $M$ is marginalized out. Similarly, $P_{XY}$ denotes the marginal distribution of $(X, Y)$.

For any fixed $d \in \mathcal{D}$, $P_{XY|D=d}$ is a joint distribution of $(X, Y)$. Note that our notation is somewhat redundant, as both $d$ and $P_{XY|D=d}$ are notations for the same thing – a domain = a joint distribution of $(X, Y)$ – but these two notations will serve different purposes in our discussion.[2]

To formalize the notion that $M$ is a partial summary of $D$, we assume that $(X, Y)$ and $M$ are conditionally independent, given $D$:

$$P_{XY|D,M} = P_{XY|D} \tag{1}$$

This implies that, given $D$, the joint distribution of $X$ and $Y$ does not change with knowledge of $M$. An important special case where this holds in when $M = g(D)$ for some deterministic $g : \mathcal{D} \to \mathcal{M}$. We illustrate this probabilistic framework with motivating examples in Table 1, whose implications will be discussed throughout the paper.

Table 1: Examples of domains and metadata in different tasks.

| Task | Input $X$ | Label $Y$ | Domain $D$ | Metadata $M$ |
|------|-----------|-----------|------------|--------------|
| Sentiment annotation (Multiple Annotators) | Sentence to be annotated | Sentiment label (e.g., positive, negative) | Annotator identity (e.g., "Annotator 1") | Annotator's demographic profile (e.g., age) |
| Review rating prediction (Multiple Reviewers) | Product review text | Numerical rating (e.g., 1–5 stars) | Reviewer identity (e.g., "Reviewer 2") | Unlabeled texts written by the reviewer $\{X_i\}_{i=1}^n \overset{iid}{\sim} P_{X|D=d}$ |
| Image classification across styles | Image | Object category label (e.g., dog, car) | Image style (e.g., photograph, sketch, painting) | Textual description of style |

The training data available to the learner is generated as follows: First, $N$ domains $d_1, \ldots, d_N$ are sampled iid from $P_D$, but not observed. Then, conditioned on these $d_i$, corresponding values $m_i$ are observed. In addition, for each $i$, $1 \le i \le N$, data $(x_{ij}, y_{ij})$ are sampled iid from $P_{XY|D=d_i}$, $1 \le j \le n_i$. In summary, the overall training data is

$$\left( m_i, (x_{ij}, y_{ij})_{j=1}^{n_i} \right)_{i=1}^N .$$

The goal of the learner is to produce a function $f$ that accurately predicts labels on a new, random domain. In particular, $f$ should minimize the risk

$$R(f) := \mathbb{E}_{X,Y,M,D} \left[ \mathbb{1}_{f(\cdot) \neq Y} \right] .$$

---

[2]Blanchard et al. [2011, 2021] use $P_{XY}$ to denote a random domain, whereas in our notation, a random domain is either $P_{XY|D}$ or just $D$. Our introduction of $D$ for a random domain allows us to use $P_{XY}$ for the "average" domain, which will be a critical concept in what follows.

In practice, this risk is estimated by holding out several of the domains, and averaging the test errors on them. This probabilistic framing of DG generalizes that of Blanchard et al. [2011, 2021]. They focus on the special case where $M$ is the marginal distribution of $X$ for the given domain, and focus on the challenges associated to learning from empirical samples of the training and test $X$-marginals.

The training setup described above naturally gives rise to two different ways of using the available data. On one hand, the learner may choose to ignore the domain information and simply pool together all training samples, treating them as if they were drawn iid from a single distribution. On the other hand, the learner may choose to leverage the observed metadata $m_i$, which serves as side information about the underlying domain. These two strategies lead to two corresponding empirical risk minimization principles. Thus, let $\mathcal{F} \subset \{\mathcal{X} \times \mathcal{M} \to \mathcal{Y}\}$ denote a class of functions that take both input feature $x$ and auxiliary metadata $m$ as input, and $\mathcal{G} \subset \{\mathcal{X} \to \mathcal{Y}\}$ a class of functions that take only $x$ as input. Consider two empirical risk minimizers:

$$\textbf{Pooling ERM: } \widehat{f}_{\text{pool}} = \arg\min_{f \in \mathcal{G}} \frac{1}{N} \sum_{i=1}^{N} \frac{1}{n_i} \sum_{j=1}^{n_i} \ell(y_{ij}, f(x_{ij})). \tag{2}$$

$$\textbf{Domain-informed (DI) ERM: } \widehat{f}_{\text{DG}} = \arg\min_{f \in \mathcal{F}} \frac{1}{N} \sum_{i=1}^{N} \frac{1}{n_i} \sum_{j=1}^{n_i} \ell(y_{ij}, f(x_{ij}, m_i)). \tag{3}$$

We are interested in when DI-ERM outperforms pooling ERM. From a theoretical perspective, we work in the large-sample and "large-model" limit (where $\mathcal{F}$ and $\mathcal{G}$ can approximate the Bayes-optimal predictor arbitrarily well). In this regime, standard learning-theoretic arguments imply that the performance of the two approaches is characterized by their corresponding Bayes risks, defined below.

## 4 Risk and Bayes risk in domain generalization

To aid in understanding domain generalization, it is helpful to consider DG in relation to two other problem settings. These settings differ only in what information the classifier $f$ has access to. In all cases, the performance measure is the risk

$$R(f) := \mathbb{E}_{X,Y,M,D} \left[ \mathbb{1}_{f(\cdot) \neq Y} \right],$$

where the argument of $f(\cdot)$ depends on settings.

**No Domain Information:** In this setting, the classifier only has access to the feature vector $x$ at test time, and is thus $f(x)$. As noted earlier, most empirical DG methods, especially in computer vision, have this form. The risk in this case is

$$R(f) = \mathbb{E}_{X,Y,M,D} \left[ \mathbb{1}_{f(X) \neq Y} \right] = \mathbb{E}_{X,Y} \left[ \mathbb{1}_{f(X) \neq Y} \right] = \mathbb{E}_X \left[ \mathbb{E}_{Y|X} \left[ \mathbb{1}_{f(X) \neq Y} \right] \right],$$

where, because $f$ does not depend on $D$ or $M$, these variables marginalize out. Therefore, the problem reduces to learning with respect to the marginal distribution of $(X, Y)$, which corresponds to pooling data across domains. The optimal classifier $f_{\text{pool}}^*$ is thus the Bayes classifier for the marginal distribution of $(X, Y)$:

$$f_{\text{pool}}^*(x) = \arg\max_k \mathbb{P}\left(Y = k | X = x\right).$$

The corresponding Bayes risk, $R_{\text{pool}}^*$, is the Bayes risk for the marginal distribution of $(X, Y)$

$$R_{\text{pool}}^* := \mathbb{E}_{X,Y} \left[ \mathbb{1}_{f_{\text{pool}}^*(X) \neq Y} \right] = \mathbb{E}_X \left[ 1 - \max_k \mathbb{P}\left(Y = k | X\right) \right].$$

**Full Domain Information:** In this setting, the classifier has full knowledge of the domain $D$ at test time, and is thus denoted $f(x, d)$. In practice, full knowledge of $D$ is not available, and this setting therefore serves as a bound on the best possible performance of DG. The risk in this setting is

$$R(f) = \mathbb{E}_{X,Y,M,D} \left[ \mathbb{1}_{f(X,D) \neq Y} \right] = \mathbb{E}_{X,Y,D} \left[ \mathbb{1}_{f(X,D) \neq Y} \right] = \mathbb{E}_{X,D} \left[ \mathbb{E}_{Y|X,D} \left[ \mathbb{1}_{f(X,D) \neq Y} \right] \right].$$

The optimal classifier $f^*_{\text{full}}$ is now the Bayes classifier for the distribution of $X, Y | D$,

$$f^*_{\text{full}}(x, d) = \arg \max_k \mathbb{P}\left(Y = k | X = x, D = d\right),$$

and the corresponding Bayes risk is:

$$R^*_{\text{full}} := \mathbb{E}_{X,Y,D}\left[\mathbb{1}_{f^*_{\text{full}}(X,D) \neq Y}\right] = \mathbb{E}_{X,D}\left[1 - \max_k \mathbb{P}\left(Y = k | X, D\right)\right].$$

In this setting, the classifier has full knowledge of the test domain, in other words, the joint distribution of $(X, Y)$ for the given test domain. Therefore, $R^*(X, D)$ is the Bayes risk for the test domain, which serves as a lower bound for the risk in domain generalization.

**Remark 1** *Achieving $R^*(X, D)$ is the goal of* domain adaptation.

**Partial Domain Information:** This is the setting of domain generalization. The classifier has access to not only $x$, but also a variable $m$ that conveys partial information about the true domain $d$. A classifier in this setting is denoted $f(x, m)$. The risk is

$$R(f) = \mathbb{E}_{X,Y,M,D}\left[\mathbb{1}_{f(X,M) \neq Y}\right] = \mathbb{E}_{X,Y,M}\left[\mathbb{1}_{f(X,M) \neq Y}\right] = \mathbb{E}_{X,M}\left[\mathbb{E}_{Y|X,M}\left[\mathbb{1}_{f(X,M) \neq Y}\right]\right].$$

The optimal classifier $f^*_{\text{DG}}$ is now the Bayes classifier for the distribution of $X, Y | M$,

$$f^*_{\text{DG}}(x, m) = \arg \max_k \mathbb{P}\left(Y = k | X = x, M = m\right),$$

and the corresponding Bayes risk is

$$R^*_{\text{DG}} := \mathbb{E}_{X,Y,M}\left[\mathbb{1}_{f^*_{\text{DG}}(X,M) \neq Y}\right] = \mathbb{E}_{X,M}\left[1 - \max_k \mathbb{P}\left(Y = k | X, M\right)\right].$$

In this setting, the optimal classifier uses both $X$ and domain-specific signal $M$ to predict $Y$. This setting aligns with the original theoretical motivations of DG and highlights the value of leveraging test-time domain information. A key goal of our work is to reassert the importance of this formulation and demonstrate both its theoretical advantages and empirical benefits, particularly in contrast to the more commonly used $f(x)$ setting.

## 5    Comparison of Bayes risks

This section develops bounds that relate the three Bayes risks defined in the previous section. The bounds reveal settings where domain information is and is not beneficial. The following basic result provides a starting point.

**Proposition 1 (Risk Hierarchy)** $R^*_{\text{pool}} \geq R^*_{\text{DG}} \geq R^*_{\text{full}}$.

The proof is straightforward (see Appendix C.1). The first inequality is trivial, as extending a feature vector can never decrease the Bayes risk. The second inequality follows from (1).

Our focus in this section is to determine conditions under which these inequalities become strict, and with a quantifiable gap. Toward that end, consider the following definition.

**Definition 1 (Point-wise Margin)** *Consider any random triple $(X, Y, M)$, where $Y$ is discrete. De-fine the point-wise margin of $Y | M = m, X = x$ as,*

$$\gamma(x, m) := \max_k \mathbb{P}\left(Y = k | X = x, M = m\right) - 2\text{nd} \max_k \mathbb{P}\left(Y = k | X = x, M = m\right).$$

The operator $2\text{nd} \max_k$ returns the second largest value of its argument. Thus, if the two largest values of $\mathbb{P}\left(Y = k | X = x, M = m\right)$ are the same, $\gamma(x, m) = 0$. Intuitively, $\gamma(x, m)$ reflects the degree of certainty that the Bayes classifier $f^*_{\text{DG}}(x, m)$ has about its prediction. The larger $\gamma(x, m)$, the more confident the prediction.

The next result gives upper and lower bounds on the gap between $R^*_{\text{DG}}$ and $R^*_{\text{pool}}$. This gap is the additional reduction in risk that results from leveraging the partial domain information $M$.

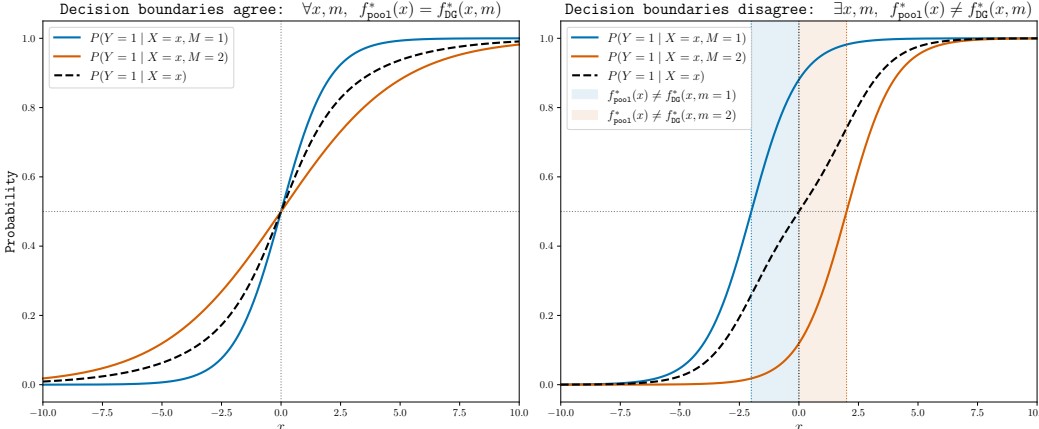

Figure 1: Illustration of Theorem 1. Consider binary classification with $X \in \mathbb{R}$, $Y \in \{1, 2\}$, and $M \in \{1, 2\}$. Then the Bayes classifiers $f^*_{\text{pool}}(x)$, $f^*_{\text{DG}}(x, m = 1)$ and $f^*_{\text{DG}}(x, m = 2)$ can be obtained by thresholding the corresponding posteriors at $1/2$. The left figure shows a scenario where the domain-informed classifier $f^*_{\text{DG}}$ and the pooled classifier $f^*_{\text{pool}}$ agree everywhere, and therefore both upper and lower bound are 0. In this case, domain information $M$ is not beneficial. The right figure shows a scenario where $f^*_{\text{DG}}$ disagrees with $f^*_{\text{pool}}$ in certain regions, and domain information does lead to lower Bayes risk.

**Theorem 1 (Risk Reduction from Domain Information)** *Consider any random triple $(X, Y, M)$, where $Y$ is discrete. Then*

$$\mathbb{E}_{X,M}\left[\gamma(X, M)\mathbb{1}_{f^*_{\text{pool}}(X) \neq f^*_{\text{DG}}(X,M)}\right] \leq R^*_{\text{pool}} - R^*_{\text{DG}} \leq \mathbb{E}_{X,M}\left[\mathbb{1}_{f^*_{\text{pool}}(X) \neq f^*_{\text{DG}}(X,M)}\right].$$

The proof of Theorem 1 is in Appendix C.2. The upper bound represents the probability of disagreement between the domain-informed classifier $f^*_{\text{DG}}$ and the pooled classifier $f^*_{\text{pool}}$. The lower bound can be interpreted as the expected cost of disagreement, where the cost is zero when the predictions agree, and equals the margin $\gamma(X, M)$ when they differ. Hence, domain information is particularly beneficial when $f^*_{\text{DG}}$ frequently disagrees with $f^*_{\text{pool}}$ in regions of high confidence. Figure 1 gives more intuition.

**Remark 2** *Although not the focus of this paper, a version of Theorem 1 also holds for the gap $R^*_{\text{DG}} - R^*_{\text{full}}$, where $R^*_{\text{full}}$ is the risk of a classifier that has full knowledge of the test domain. This bound quantifies the difference between DG and domain adaptation, and addresses a question left open by Blanchard et al. [2021, Lemma 9], see Appendix B for more detailed discussion.*

## 5.1 No information-theoretic gain under covariate shift

Theorem 1 holds regardless of the distribution $P_{XYMD}$. By considering assumptions on this distribution, stronger conclusions may be drawn. Covariate shift refers to the setting where, as the domain $D$ varies, $P_{X|D}$ changes, but $P_{Y|X,D}$ does not. More generally, we can extend the meaning of covariate shift to be any DG problem where the Bayes classifier $f^*_{\text{DG}}$ does not depend on $M$. In such a scenario, domain-specific information is of no benefit.

**Corollary 1** *Under covariate shift, $R^*_{\text{pool}} = R^*_{\text{DG}}$.*

## 5.2 Information-theoretic gain under posterior drift

We now examine a class of distributions where the gap $R^*_{\text{pool}} - R^*_{\text{DG}}$ has a more concrete lower bound. This class of distributions is motivated by applications—particularly in natural language processing—where the posterior $P_{Y|X,D}$ differs across domains due to inherent ambiguity or subjectivity. A canonical example is the sentiment or toxicity annotation task, where annotators often disagree

on the same text. For instance, in the age-related sentiment analysis dataset of Díaz et al. [2018], the sentence *"Old people's appearance contains so much lived life."* received conflicting labels: 2/5 annotators seeing it as 'very positive', 2/5 as 'somewhat positive', and 1/5 as 'very negative'. This reflects how labeling tendency varies with annotator identity. We capture this phenomenon by introducing a formal posterior drift class for domain generalization.

**Definition 2 (Posterior Drift Class for Domain Generalization)**

$$\Pi(\gamma, \epsilon) := \Big\{ (X, Y, M, D) : \forall x, m, \ \gamma(x, m) \geq \gamma, \ and$$

$$P_{X, M, M'}\Big( f^*_{\mathrm{DG}}(X, M) \neq f^*_{\mathrm{DG}}(X, M') \Big) \geq \epsilon \Big\},$$

*where $(M, M') \mid X \sim P_{M|X} \otimes P_{M|X}$ are two independent draws.*

This class of DG problems captures settings where optimal classifiers with different $M$ make conflicting predictions on a non-negligible region of the input space. The parameter $\gamma$ quantifies the point-wise confidence of the optimal predictor, the parameter $\epsilon$ quantifies the average amount of variation in $P_{Y|X,M}$ for different $M$. With this, we have an explicit lower bound:

**Proposition 2**

$$\inf_{(X,Y,M) \in \Pi(\gamma, \epsilon)} \Big[ R^*_{\mathrm{pool}} - R^*_{\mathrm{DG}} \Big] \geq \frac{\gamma \cdot \epsilon}{2}$$

The proof is in Appendix C.3. This lower bound shows that leveraging domain-specific information yields a provable benefit of at least $\gamma \epsilon / 2$ for this particular formulation of posterior drift.

In contrast to pessimistic results in *domain adaptation*—where no method consistently outperforms vanilla ERM under posterior drift [Zhu et al., 2024, Liu et al., 2024]—our work presents an optimistic view in *domain generalization*: by conditioning on domain metadata $M$, we can provably do better than pooling-based prediction.

## 5.3 Advantage of DI-ERM beyond posterior drift

The previous subsection demonstrates the information-theoretic gain from incorporating domain information $M$. We now consider the practical setting of learning under restricted function classes.

Let $\mathcal{F} \subset \{\mathcal{X} \times \mathcal{M} \to \mathcal{Y}\}$ denote a class of predictors that take both input features $x$ and auxiliary metadata $m$ as input. From this class, we define a corresponding class $\mathcal{G} \subset \{\mathcal{X} \to \mathcal{Y}\}$ as:

$$\mathcal{G} := \{x \mapsto f(x, m_0) : f \in \mathcal{F}, \ m_0 \in \mathcal{M}\},$$

i.e., $\mathcal{G}$ consists of classifiers in $\mathcal{F}$ where the metadata variable $m$ is held fixed.

Clearly, any function in $\mathcal{G}$ is realizable within $\mathcal{F}$. Therefore,

$$R^*_{\mathrm{pool}, \mathcal{G}} := \inf_{f \in \mathcal{G}} R(f) \quad \geq \quad R^*_{\mathrm{DG}, \mathcal{F}} := \inf_{f \in \mathcal{F}} R(f). \tag{4}$$

We are interested in understanding when strict inequality holds. The example below show that even if there is no information-theoretic gain of DI-ERM under covariate shift (Corollary 1), it may still have practical advantage when considering a restricted function class $\mathcal{F}$.

**Example 1 (Covariate shift without posterior drift)** *Let $P_{XYM}$ be*

$$M \sim \mathrm{Bernoulli}(p), \ where \ p > 1/2, \quad \begin{cases} M = 1: & X \sim \mathrm{Unif}[0, 2], \ Y = \mathrm{sign}(X - 1) \\ M = 2: & X \sim \mathrm{Unif}[4, 6], \ Y = \mathrm{sign}(X - 5). \end{cases}$$

*Because the supports are disjoint, the pooling and DG Bayes classifier are the same, to be specific*

$$f^*_{\mathrm{pool}}(x) = \begin{cases} \mathrm{sign}(x - 1), & x \in [0, 2] \\ \mathrm{sign}(x - 5), & x \in [4, 6] \end{cases}, \quad f^*_{\mathrm{DG}}(x, m) = \mathrm{sign}(x - 4m + 3) \quad \Longrightarrow f^*_{\mathrm{pool}} = f^*_{\mathrm{DG}}$$

therefore $R^*_{\text{pool}} = R^*_{\text{DG}} = 0$. *The model classes are linear classifiers*

$$\mathcal{F} = \big\{ f(x, m) = \text{sign}(w^\top x + v^\top m + b) \big\}, \qquad \mathcal{G} = \big\{ f(x) = \text{sign}(w^\top x + b) \big\}.$$

*$\mathcal{F}$ can realize $f^*_{\text{DG}}$ with a bias term that depends on $m$, giving $R^*_{\text{DG}, \mathcal{F}} = 0$. However, a predictor in $\mathcal{G}$ can only choose a single threshold, and the optimal one is*

$$f^*_{\text{pool}, \mathcal{G}}(x) = \text{sign}(x - 1), \quad R^*_{\text{pool}, \mathcal{G}} = \frac{\min\{p, 1 - p\}}{2}.$$

*Therefore, $R^*_{\text{pool}, \mathcal{G}} > R^*_{\text{DG}, \mathcal{F}}$, even though $R^*_{\text{pool}} = R^*_{\text{DG}}$.*

This toy construction mirrors image classification task across different styles: each style (domain) has a separate support, so the Bayes classifier is the same with or without $m$, yet $m$ still helps within a restricted model class. This is experimentally verified in Section 6 .

# 6 Experiments

We evaluate the effectiveness of domain-informed ERM (DI-ERM) in three experimental settings. Our primary focus is on the comparison between DI-ERM and pooling ERM, which highlights the benefit of incorporating domain metadata. Additional results—including linear probing, benchmarks against alternative methods, and complete experimental details—are provided in Appendix D.

**Sentiment disagreement among annotators** In many NLP tasks, annotators exhibit subjective preferences, leading to disagreement of the label $y$ on the same input $x$—a form of posterior drift discussed in Section 5.2. To study this phenomenon, we use the dataset of Díaz et al. [2018], which re-annotates a subset of Sentiment140 for training and provides a test set drawn from age-related blog posts. The training set comprises 59,235 sentences labeled by 1,481 annotators; the test set includes 1,419 sentences labeled by 878 annotators. Each sentence is annotated by 4–5 individuals, and the labels exhibit high disagreement (about 40 %). In this setting, the input $x$ is a sentence, the label $y \in \{1, 2, 3, 4, 5\}$ denotes sentiment on a five-point scale, the domain $d$ corresponds to the annotator, and the domain information $m$ consists of demographic metadata (e.g., age, upbringing region).

To encode domain information $M$, we concatenate it with the sentence $x$ in a text-prompt format, as illustrated in Figure 2. Table 3 reports the results. DI-ERM substantially outperforms pooling ERM, demonstrating that leveraging annotator metadata can dramatically improve predictive accuracy. Notably, DI-ERM also surpasses the previous state-of-the-art reported by Deng et al. [2023].

Table 3: Test accuracy on the sentiment disagreement dataset. Incorporating annotator profiles ($M$) through DI-ERM yields a dramatic improvement over pooling ERM, reflecting the importance of modeling annotator-specific posterior drift. In particular, DI-ERM nearly doubles accuracy compared to pooling ERM and surpasses the previous state-of-the-art (69.8% by [Deng et al., 2023]).

| Algorithm | Model | Test Avg Acc |
|---|---|---|
| Pooling ERM (finetune) | BERT | $49.1 \pm 0.4$ |
| DI-ERM (finetune) | BERT | $90.5 \pm 0.2$ |

**Reviewer-specific sentiment analysis** We next examine the WILDS-Amazon Reviews dataset [Koh et al., 2021], which captures distributional shifts across reviewers. Here, the input $x$ is a product review, $y \in \{1, \ldots, 5\}$ is the star rating, $d$ denotes the reviewer identity, and $m$ consists of all (unlabeled) reviews written by that reviewer.

The central hypothesis is that a reviewer's writing style $M = P_{X|D}$ provides a useful signal for predicting their rating behavior $P_{Y|X,D}$. The training set contains 245,502 reviews from 1,252 reviewers, while the test set consists of 100,050 reviews from 1,334 unseen reviewers.

To incorporate reviewer context $M$, we randomly sample 20 additional reviews written by the same reviewer and concatenate them with the current review in a prompt format, shown in Figure 3. As summarized in Table 4, DI-ERM outperforms pooling ERM. Beyond higher average accuracy, DI-ERM also boosts the 10th-percentile accuracy across reviewers—a key robustness metric used on the official leaderboard. With end-to-end fine-tuning, DI-ERM surpassing the best leaderboard result on both metrics (see Appendix D for additional discussion).

Table 4: Sentiment classification performance on Amazon-WILDS with reviewer-specific context. DI-ERM consistently improves over pooling ERM, both in average accuracy and in 10th-percentile reviewer accuracy—the official leaderboard metric. It also exceeds the best result reported on the WILDS leaderboard (`https://wilds.stanford.edu/`).

| Algorithm | Model | Test Avg Acc | Test 10% Acc |
|---|---|---|---|
| Pooling ERM (finetune) | nomic-embed-text-v1.5 | $71.8 \pm 0.9$ | $54.7 \pm 0.0$ |
| DI-ERM (finetune) | nomic-embed-text-v1.5 | $73.1 \pm 0.3$ | $56.4 \pm 0.8$ |

**Image classification across styles** We next evaluate our method on the PACS dataset [Li et al., 2017], which contains images drawn from four distinct visual styles: $d \in$ Photo (P), Art Painting (A), Cartoon (C), Sketch (S). Each image $x$ belongs to one of seven categories, $y \in \{$Dog, Elephant, Giraffe, Guitar, Horse, House, Person$\}$. Domain information is represented by a short text description $m$, such as "a photo" or "a pencil sketch" (see Figure 4).

This vision task satisfies covariate shift, since a single classifier should accurately classify all images across domains. Thus, in line with Section 5.3, we expect any gains to be due to using a restricted function class. To implement DI-ERM, we use pretrained image foundation models (e.g., CLIP [Radford et al., 2021], DINOv2 [Oquab et al., 2023]) to extract visual features from $x$, and encode the domain description $m$ using a pretrained language model (DistilBERT) following the prompt in Figure 4. The resulting image and text embeddings are concatenated into a joint representation for classification.

We follow the standard PACS evaluation protocol: training on three domains and testing on the held-out fourth domain, repeated across all domain splits. All encoders are frozen, and linear classifiers are trained on top of the fixed representations.

As shown in Table 10, DI-ERM improves over pooling ERM in most settings. The gains are most pronounced for mid-sized models, while the benefit diminishes for larger foundation models. This pattern aligns with the discussion in Section 5.3: under covariate shift, the benefit of DI-ERM decreases as model mismatch becomes smaller.

Table 5: Domain generalization results on PACS using models from the CLIP and DINOv2 families. DI-ERM achieves improved accuracy over pooling ERM in most configurations, particularly for mid-sized models, less noticeable for large-sized models. When using large-sized models, both ERM and DI-ERM approaches SOTA performance.

| Algorithm | Model | PAC $\rightarrow$ S | ACS $\rightarrow$ P | CSP $\rightarrow$ A | SPA $\rightarrow$ C | Test Avg Acc |
|---|---|---|---|---|---|---|
| Pooling ERM (linear) | CLIP: vitb32 | 86.97 | 99.58 | 95.90 | 97.48 | 94.98 |
| DI-ERM (linear) | | **88.06** | **99.64** | **96.29** | **97.48** | **95.37** |
| Pooling ERM (linear) | CLIP: vitl14 | 95.42 | 99.94 | 99.22 | 99.79 | 98.59 |
| DI-ERM (linear) | | 95.32 | 99.94 | **99.32** | 99.79 | 98.59 |
| Pooling ERM (linear) | DINOv2: vits14 | 79.82 | 85.81 | 93.55 | 91.34 | 87.63 |
| DI-ERM (linear) | | **80.45** | **90.00** | **94.09** | **91.60** | **89.04** |
| Pooling ERM (linear) | DINOv2: vitl14 | 92.29 | 96.41 | 98.14 | 97.48 | 96.08 |
| DI-ERM (linear) | | **92.42** | **97.37** | 98.10 | 97.48 | **96.34** |

# 7 Conclusions

This work presents a rigorous theory of domain generalization, precisely characterizing when and why leveraging domain information at test time is beneficial. Empirically, we demonstrate that domain-informed ERM (DI-ERM) outperforms pooled ERM across three representative scenarios in language and vision tasks. Future work can be done to explore alternative ways of encoding domain information, and a broader range of DG benchmarks.

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

# A   Appendix: additional literature review

**ERM is hard to beat.**   Empirically, Gulrajani and Lopez-Paz [2021] first emphasized that a well-tuned empirical risk minimization (ERM) baseline outperforms many domain generalization (DG) methods on vision benchmarks. Similar patterns were later observed on the WILDS benchmark [Koh et al., 2021], and again in the context of federated learning by Bai et al. [2024]. On the theoretical front, Rosenfeld et al. [2021] and Gouk et al. [2024] studied function classes of the form $f : \mathcal{X} \to \mathcal{Y}$ and concluded that, under common assumptions, ERM cannot be fundamentally outperformed (e.g., in terms of minimax risk).

**The use of unlabeled data.**   While most DG methods restrict themselves to using only the input $x$ at inference time, some methods explore the use of unlabeled test-domain data. Several DG methods attempt to exploit unlabeled test data to improve generalization [Blanchard et al., 2011, Muandet et al., 2013, Zhang et al., 2021]. A closely related setting is unsupervised domain adaptation (UDA), where unlabeled test data are used to adapt models to the test domain. Unlike DG, UDA assumes access to target-domain data at training time and typically requires learning a separate model per test domain [Sun and Saenko, 2016, Ganin et al., 2016].

Although promising in principle, the practical benefits of using unlabeled data remain mixed. A large-scale study by Sagawa et al. [2022] evaluating methods across ten diverse datasets found that incorporating unlabeled data frequently failed to improve upon strong ERM baselines. These findings reinforce the need for a more precise understanding of when and how unlabeled data can contribute to domain generalization.

Our framework addresses this gap by casting unlabeled data as a special case of auxiliary domain information, and by providing conditions under which such information is expected to improve generalization performance.

# B   Appendix: Partial versus full domain knowledge

Although not the focus of this paper, a version of Theorem 1 also holds for the gap $R^*_{\mathrm{DG}} - R^*_{\mathrm{full}}$, where $R^*_{\mathrm{full}}$ is the risk of a classifier that has full knowledge of the test domain. Such a bound addresses a question left open by Blanchard et al. [2021, Lemma 9], who established that this gap is lower bounded by zero, and provide a condition under which the gap equals zero. The following result bounds this gap in a more general setting.

**Proposition 3** *Let*

$$\widetilde{\gamma}(x, d) := \max_k \mathbb{P}\left(Y = k | X = x, D = d\right) - 2\mathrm{nd} \max_k \mathbb{P}\left(Y = k | X = x, D = d\right).$$

*Then*

$$\mathbb{E}_{X,D,M}\left[\widetilde{\gamma}(X, D)\mathbb{1}_{f^*_{\mathrm{full}}(X,D) \neq f^*_{\mathrm{DG}}(X,M)}\right] \leq R^*_{\mathrm{DG}} - R^*_{\mathrm{full}} \leq \mathbb{E}_{X,D,M}\left[\mathbb{1}_{f^*_{\mathrm{full}}(X,D) \neq f^*_{\mathrm{DG}}(X,M)}\right]$$

*and in particular,*

$$f^*_{\mathrm{full}}(x, d) = f^*_{\mathrm{DG}}(x, m) \quad \textit{almost surely w.r.t. } P_{XMD} \implies R^*_{\mathrm{full}} = R^*_{\mathrm{DG}}.$$

The result has an interpretation analogous to that of Theorem 1. In particular, if the observed domain information is of low quality, in the sense that $f^*_{\mathrm{full}}$ disagrees with $f^*_{\mathrm{DG}}$ often, and with high confidence, then $R^*_{\mathrm{DG}}$ can be substantially worse than $R^*_{\mathrm{full}}$.

# C   Appendix: proofs

## C.1   Proof of Proposition 1

**Proposition (Risk Hierarchy)**   $R^*_{\mathrm{pool}} \geq R^*_{\mathrm{DG}} \geq R^*_{\mathrm{full}}$.

*Proof.*

$$R^*_{\text{pool}} = \inf_{f:\mathcal{X}\to\mathcal{Y}} \mathbb{E}_{X,Y,M,D}\left[\mathbb{1}_{f(X)\neq Y}\right]$$
$$\geq \inf_{f:\mathcal{X}\times\mathcal{M}\to\mathcal{Y}} \mathbb{E}_{X,Y,M,D}\left[\mathbb{1}_{f(X,M)\neq Y}\right] = R^*_{\text{DG}}$$
$$\geq \inf_{f:\mathcal{X}\times\mathcal{M}\times\mathcal{D}\to\mathcal{Y}} \mathbb{E}_{X,Y,M,D}\left[\mathbb{1}_{f(X,M,D)\neq Y}\right]$$
$$= \inf_{f:\mathcal{X}\times\mathcal{D}\to\mathcal{Y}} \mathbb{E}_{X,Y,M,D}\left[\mathbb{1}_{f(X,D)\neq Y}\right] = R^*_{\text{full}} \qquad \because (X,Y)|M,D = (X,Y)|D$$

∎

## C.2 Proof of Theorem 1

**Theorem (Risk Reduction from Domain Information)** *Consider any random triple $(X,Y,M)$, where $Y$ is discrete. Then*

$$\mathbb{E}_{X,M}\left[\gamma(X,M)\mathbb{1}_{f^*_{\text{pool}}(X)\neq f^*_{\text{DG}}(X,M)}\right] \leq R^*_{\text{pool}} - R^*_{\text{DG}} \leq \mathbb{E}_{X,M}\left[\mathbb{1}_{f^*_{\text{pool}}(X)\neq f^*_{\text{DG}}(X,M)}\right].$$

*Proof.* The gap in the two risks can be expressed as

$$R^*_{\text{pool}} - R^*_{\text{DG}}$$
$$= \mathbb{E}_X\left[\mathbb{E}_{M|X}\left[\mathbb{P}\left(Y = f^*_{\text{DG}}(X,M)|X,M\right)\right]\right] - \mathbb{E}_X\left[\mathbb{P}\left(Y = f^*_{\text{pool}}(X)|X\right)\right]$$
$$= \mathbb{E}_X\left[\mathbb{E}_{M|X}\left[\mathbb{P}\left(Y = f^*_{\text{DG}}(X,M)|X,M\right)\right]\right] - \mathbb{E}_X\left[\mathbb{E}_{M|X}\left[\mathbb{P}\left(Y = f^*_{\text{pool}}(X)|X,M\right)\right]\right]$$
$$= \mathbb{E}_X\left[\mathbb{E}_{M|X}\left[\mathbb{P}\left(Y = f^*_{\text{DG}}(X,M)|X,M\right) - \mathbb{P}\left(Y = f^*_{\text{pool}}(X)|X,M\right)\right]\right]$$

Notice that for any $x, m$, if $f^*_{\text{DG}}(x,m) = f^*_{\text{pool}}(x)$, then the pointwise difference of the conditional probabilities inside the expectation above must be zero.

Whereas if they disagree, then it must hold that

$$P(Y = f^*_{\text{pool}}(X)|X,M) \leq \text{2nd}\max_k P(Y = k|X,M)$$

due to the definition of $f^*_{\text{DG}}$.

It thus follows that.

$$\gamma(x,m)\mathbb{1}_{f^*_{\text{pool}}(x)\neq f^*_{\text{DG}}(x,m)} \leq \mathbb{P}\left(Y = f^*_{\text{DG}}(x,m)|X=x, M=m\right) - \mathbb{P}\left(Y = f^*_{\text{pool}}(x)|X=x, M=m\right)$$
$$\leq \mathbb{1}_{f^*_{\text{pool}}(x)\neq f^*_{\text{DG}}(x,m)}.$$

The inequalities in the theorem statement now follow.

∎

## C.3 Proof of Proposition 2

**Proposition**

$$\inf_{(X,Y,M)\in\Pi(\gamma,\epsilon)}\left[R^*_{\text{pool}} - R^*_{\text{DG}}\right] \geq \frac{\gamma\cdot\epsilon}{2}$$

*Proof.* From the lower bound in Theorem 1, we have

$$R^*_{\text{pool}} - R^*_{\text{DG}} \geq \mathbb{E}_{X,M}\left[\gamma(X,M)\mathbb{1}_{f^*_{\text{pool}}(X)\neq f^*_{\text{DG}}(X,M)}\right] \qquad \because \text{ Theorem 1}$$
$$\geq \gamma\,\mathbb{E}_{X,M}\left[\mathbb{1}_{f^*_{\text{pool}}(X)\neq f^*_{\text{DG}}(X,M)}\right] \qquad \because \text{ margin assumption in } \Pi(\gamma,\epsilon)$$
$$= \gamma\,\mathbb{E}_X\left[\mathbb{E}_{M|X}\left[\mathbb{1}_{f^*_{\text{pool}}(X)\neq f^*_{\text{DG}}(X,M)}\right]\right]$$

Now we will show that

$$\forall x, \quad \mathbb{E}_{M|X=x}\left[\mathbb{1}_{f^*_{\text{pool}}(x)\neq f^*_{\text{DG}}(x,M)}\right] \geq \frac{1}{2}\,\mathbb{E}_{M,M'|X=x}\left[\mathbb{1}_{f^*_{\text{DG}}(x,M)\neq f^*_{\text{DG}}(x,M')}\right],$$

where

$$M,\ M'\overset{\text{i.i.d.}}{\sim} P_{M|X=x}.$$

Let's examine the two terms. Fix $x$, denote

$$\pi_k(x) = \mathbb{P}\left(f^*_{\text{DG}}(x,M)=k|X=x\right),$$

note that the randomness comes from $M$.

Then for any $x$,

$$
\begin{aligned}
\mathbb{E}_{M,M'|X=x}\left[\mathbb{1}_{f^*_{\text{DG}}(x,M)\neq f^*_{\text{DG}}(x,M')}\right] &= \mathbb{P}\left(f^*_{\text{DG}}(x,M)\neq f^*_{\text{DG}}(x,M')|X=x\right)\\
&= \sum_k \mathbb{P}\left(f^*_{\text{DG}}(x,M)=k, f^*_{\text{DG}}(x,M')\neq k|X=x\right)\\
&= \sum_k \pi_k(x)\left(1-\pi_k(x)\right)\\
&= 1-\sum_k \pi_k(x)^2.
\end{aligned}
$$

Now assume $f^*_{\text{pool}}(x)=k_0$, then

$$
\begin{aligned}
\mathbb{E}_{M|X=x}\left[\mathbb{1}_{f^*_{\text{pool}}(x)\neq f^*_{\text{DG}}(x,M)}\right] &= \mathbb{E}_{M|X=x}\left[\mathbb{1}_{f^*_{\text{DG}}(x,M)\neq k_0}\right]\\
&= 1-\pi_{k_0}(x)
\end{aligned}
$$

Notice that

$$
\begin{aligned}
1-\sum_k \pi_k(x)^2 &\leq 1-\pi_{k_0}^2 && \text{``=" \textbf{when} } \pi_{k_0}=1\\
&= (1+\pi_{k_0})(1-\pi_{k_0})\\
&\leq 2(1-\pi_{k_0}) && \text{``=" \textbf{when} } \pi_{k_0}=1.
\end{aligned}
$$

Then

$$\mathbb{E}_{M|X=x}\left[\mathbb{1}_{f^*_{\text{pool}}(x)\neq f^*_{\text{DG}}(x,M)}\right] \geq \frac{1}{2}\,\mathbb{E}_{M,M'|X=x}\left[\mathbb{1}_{f^*_{\text{DG}}(x,M)\neq f^*_{\text{DG}}(x,M')}\right].$$

Integrate over $x$, we have

$$
\begin{aligned}
\mathbb{E}_X\left[\mathbb{E}_{M|X}\left[\mathbb{1}_{f^*_{\text{pool}}(x)\neq f^*_{\text{DG}}(X,M)}\right]\right] &\geq \frac{1}{2}\mathbb{E}_X\left[\mathbb{E}_{M,M'|X}\left[\mathbb{1}_{f^*_{\text{DG}}(X,M)\neq f^*_{\text{DG}}(X,M')}\right]\right]\\
&= \frac{1}{2}P_{X,M,M'}\left(f^*_{\text{DG}}(X,M)\neq f^*_{\text{DG}}(X,M')\right)\\
&\geq \frac{1}{2}\epsilon && \because \text{by definition of } \Pi(\gamma,\epsilon)
\end{aligned}
$$

$\blacksquare$

## C.4   Proof of Proposition 3

**Proposition**   *Let*

$$\widetilde{\gamma}(x,d) := \max_k \mathbb{P}\left(Y=k|X=x,D=d\right) - \text{2nd}\max_k \mathbb{P}\left(Y=k|X=x,D=d\right)$$

*Then*

$$\mathbb{E}_{X,D,M}\left[\gamma(X,D)\mathbb{1}_{f^*_{\text{full}}(X,D)\neq f^*_{\text{DG}}(X,M)}\right] \leq R^*_{\text{DG}} - R^*_{\text{full}} \leq \mathbb{E}_{X,D,M}\left[\mathbb{1}_{f^*_{\text{full}}(X,D)\neq f^*_{\text{DG}}(X,M)}\right]$$

*and in particular,*

$$f_{\text{full}}^*(x, d) = f_{\text{DG}}^*(x, m) \quad \textit{almost surely w.r.t. } P_{XMD} \implies R_{\text{full}}^* = R_{\text{DG}}^*.$$

*Proof.*

$$R_{\text{DG}}^* - R_{\text{full}}^* = \mathbb{E}_{X,Y,D,M}\left[\mathbb{1}_{Y \neq f_{\text{full}}^*(X,D)}\right] - \mathbb{E}_{X,Y,D,M}\left[\mathbb{1}_{Y \neq f_{\text{DG}}^*(X,M)}\right]$$
$$= \mathbb{E}_{X,D,M}\left[\mathbb{P}\left(Y = f_{\text{DG}}^*(X,M)\right) - \mathbb{P}\left(Y = f_{\text{full}}^*(X,D)\right) | X, D, M\right]$$

Recall the assumption on $M$:

$$Y|X, D, M = Y|X, D.$$

Then for every $x$, $d$ and $m$,

$$\mathbb{P}\left(Y = f_{\text{DG}}^*(x, m) | X = x, D = d, M = m\right) - \mathbb{P}\left(Y = f_{\text{full}}^*(x, d) | X = x, D = d, M = m\right)$$
$$= \mathbb{P}\left(Y = f_{\text{DG}}^*(x, m) | X = x, D = d\right) - \mathbb{P}\left(Y = f_{\text{full}}^*(x, d) | X = x, D = d\right)$$
$$\geq \gamma(x, d) \mathbb{1}_{f^*(x,m) \neq f^*(x,d)}.$$

Similarly,

$$\mathbb{P}\left(Y = f_{\text{DG}}^*(x, m) | X = x, D = d, M = m\right) - \mathbb{P}\left(Y = f_{\text{full}}^*(x, d) | X = x, D = d, M = m\right)$$
$$\leq \mathbb{1}_{f^*(x,m) \neq f^*(x,d)}.$$

Integrate over $x, d, m$, we get the lower and upper bound.

From the lower and upper bound, we can directly get the sufficient condition

$$f_{\text{full}}^*(x, d) = f_{\text{DG}}^*(x, m) \quad \text{almost surely w.r.t. } P_{XMD}. \implies R_{\text{full}}^* = R_{\text{DG}}^*$$

∎

# D   Appendix: Experimental details

This section provides additional details on our experimental setup, models, and performance comparisons. Unless otherwise specified, all models used for fine-tuning are implemented using publicly available checkpoints (e.g., via Huggingface, Pytorch, or official Github repo). For linear probing experiments, we extract feature representations using pre-trained transformers and train downstream classifiers with `scikit-learn`, using either logistic regression or multilayer perceptrons (MLPs).

The following subsections follows the same structure as Section 6, while providing additional details and full tables.

## D.1   Sentiment disagreement among annotators

**Fine-Tuning.**   We fine-tune the `bert-base-uncased` model and benchmark DI-ERM against other domain generalization methods. For DI-ERM, we concatenate the sentence $x$ with the annotator profile $m$ using the text prompt shown in Figure 2.

Table 6 reports the results. Our models consistently outperform prior work, with the best configuration achieving over $90\%$ test accuracy—substantially higher than the previous state-of-the-art reported by Deng et al. [2023].

**Linear/MLP-probing.**   We also evaluate in a frozen-feature setting, where the language model is fixed and a lightweight classifier is trained on top. Here, $x$ is encoded with a pretrained sentiment model (e.g., [CLS] embedding of DistilBERT fine-tuned on SST-2), while $m$ is encoded with a general-purpose DistilBERT. The embeddings are concatenated and passed to either a linear or shallow MLP classifier. The classifiers are trained in `scikit-learn`.

Table 7 presents the results. DI-ERM consistently outperforms pooling ERM across different feature extractors.

```
Instruction:  Read the following sentence and the annotator's
demographic profile and determine how positive or negative the
annotator judged the sentence on a 1-5 scale (1 = Very negative, 5
= Very positive).

Sentence:  [sentence goes here]

Annotator profile:  Age {age}, Race {race}, Hispanic/Latino {hisp},
grew up in {grew}, currently lives in {curr}, region {region}, income
{income}, education {education}, employment {employment}, living
situation {living}, politics {politics}, gender {gender}.

Answer:
```

Figure 2: Text prompt that encodes annotator profile.

Table 6: Test accuracy on the sentiment disagreement dataset (fine-tuning BERT). DI-ERM (ours) achieves the best performance.

| Algorithm | Model | Test Avg Acc |
|---|---|---|
| ERM | BERT | $49.1 \pm 0.4$ |
| IRM | BERT | $48.1 \pm 0.7$ |
| GroupDRO | BERT | $49.1 \pm 0.1$ |
| CORAL | BERT | $48.4 \pm 0.2$ |
| AnnEmb (SOTA) | BERT | $64.6 \pm 0.8$ |
| DI-ERM (ours, fine-tune) | BERT | $\mathbf{90.5 \pm 0.2}$ |

Table 7: Test accuracy on the sentiment disagreement dataset (frozen feature extractor). DI-ERM consistently outperforms pooling ERM, and in some settings surpasses the prior state-of-the-art of Deng et al. [2023]. We highlight the best performance reported by Deng et al. [2023] (69.77) and our highest score (83.41). †: Checkpoints used in Deng et al. [2023] were not publicly specified.

| Algorithm | Model | Test Avg Acc |
|---|---|---|
| | BERT[†] | 64.61 |
| Deng et al. [2023] | RoBERTa[†] | 60.30 |
| | DeBERTa[†] | 69.77 |
| Pooling ERM (linear) | distilbert-base-uncased-finetuned-sst-2-english | 45.85 |
| DI-ERM (linear) | distilbert-base-uncased-finetuned-sst-2-english | **46.42** |
| Pooling ERM (MLP) | distilbert-base-uncased-finetuned-sst-2-english | 55.07 |
| DI-ERM (MLP) | distilbert-base-uncased-finetuned-sst-2-english | **78.45** |
| Pooling ERM (linear) | bert-base-multilingual-uncased-sentiment | 43.06 |
| DI-ERM (linear) | bert-base-multilingual-uncased-sentiment | **43.94** |
| Pooling ERM (MLP) | bert-base-multilingual-uncased-sentiment | 53.90 |
| DI-ERM (MLP) | bert-base-multilingual-uncased-sentiment | **83.41** |

### D.2 Reviewer-specific sentiment analysis

**Fine-Tuning.** We fine-tune the `bert-base-uncased` model and benchmark DI-ERM against other domain generalization methods. For DI-ERM, we concatenate each review $x$ with reviewer context $m$, represented by 20 randomly selected reviews from the same reviewer, using the text prompt in Figure 3.

We choose `nomic-embed-text-v1.5`, which supports a 2048-token window (compared to 512 for DistilBERT), in order to handle the long reviwer context $m$.

Table 8 reports the results. DI-ERM achieves the best performance, outperforming previously reported methods on the WILDS leaderboard (`https://wilds.stanford.edu/`).

```
Instruction:  Classify the current review based on this
reviewer's sentiment patterns.

Current Review:  [current review goes here]

Reviewer's Historical Reviews:
Review 1:  [review_1]  |  Review 2:  [review_2]  |  ...
```

Figure 3: Text prompt that encodes reviewer writing style

Table 8: Reviewer-specific sentiment analysis. DI-ERM (ours) achieves the highest accuracy, outperforming prior state-of-the-art.

| Algorithm | Model | Test Avg Acc | Test 10% Acc |
|---|---|---|---|
| ERM | DistilBERT | $72.0 \pm 0.1$ | $54.2 \pm 0.8$ |
| GroupDRO | DistilBERT | $70.0 \pm 0.5$ | $53.3 \pm 0.8$ |
| CORAL | DistilBERT | $71.1 \pm 0.3$ | $52.9 \pm 0.8$ |
| IRM | DistilBERT | $70.3 \pm 0.6$ | $52.4 \pm 0.8$ |
| LISA (SOTA) | DistilBERT | $70.7 \pm 0.3$ | $54.7 \pm 0.0$ |
| ERM (finetune) | nomic-embed-text-v1.5 | $71.8 \pm 0.9$ | $54.7 \pm 0.0$ |
| DI-ERM (ours, finetune) | nomic-embed-text-v1.5 | $73.1 \pm 0.3$ | $56.4 \pm 0.8$ |

**Linear/MLP-probing.** We also evaluate in a frozen-feature setting, where the language model is fixed and only a lightweight classifier is trained. Each review $x$ is represented by its [CLS] embedding from a pretrained sentiment model (e.g., DistilBERT fine-tuned on SST-2). For reviewer context $m$, we average the [CLS] embeddings of all reviews written by that reviewer. The concatenated review and reviewer embeddings are then passed to a linear or a shallow MLP classifier implemented in `scikit-learn`.

**Domain2Vec.** Inspired by Zaheer et al. [2017], Deshmukh et al. [2018], we implement a Domain2Vec-style module to encode reviewer-specific domain information. Given a set of reviews $\{x_1, x_2, \ldots, x_n\} \sim P_{X|D=d}$ written by reviewer $d$, we learn a mapping

$$f(\{x_1, x_2, \ldots, x_n\}) = \rho \left( \frac{1}{n} \sum_{i=1}^{n} \phi(x_i) \right),$$

where $\phi$ and $\rho$ are MLPs that map individual feature representations (extracted from pretrained model) to a latent space and then transform the aggregated feature, respectively. The resulting vector is concatenated with the review representation $x$ to predict its sentiment label $y$.

Table 9 shows the result.

### D.3 Image classification across styles

We evaluate our approach on the PACS benchmark, which contains four visual styles: Photo (P), Art Painting (A), Cartoon (C), and Sketch (S). To assess robustness to style variation, we test a diverse set of models from the CLIP and DINOv2 families.

For all the experiment we use the text prompt in Figure 4 as input to `DistillBERT`.

Table 10 summarizes the results. Across most domain shifts, our proposed DI-ERM method consistently outperforms standard pooling ERM, highlighting the advantage of incorporating domain-specific information into the representation.

Table 9: Sentiment classification on Amazon-WILDS with reviewer-specific signals. "Domain2Vec" denotes reviewer encoding based on a learned mean embedding. DI-ERM variants consistently outperform pooling ERM baselines.

| Algorithm | Model | Test Avg Acc | Test 10% Acc |
|---|---|---|---|
| Pooling ERM (linear) | distilbert-base-uncased-finetuned-sst-2-english | 67.42 | 48.00 |
| DI-ERM (linear) | distilbert-base-uncased-finetuned-sst-2-english | **68.21** | 48.00 |
| Pooling ERM (MLP) | distilbert-base-uncased-finetuned-sst-2-english | 67.59 | 48.00 |
| DI-ERM (MLP) | distilbert-base-uncased-finetuned-sst-2-english | **68.28** | **49.33** |
| DI-ERM (Domain2Vec) | distilbert-base-uncased-finetuned-sst-2-english | 68.21 | 48.00 |
| Pooling ERM (linear) | bert-base-multilingual-uncased-sentiment | 72.14 | 53.33 |
| DI-ERM (linear) | bert-base-multilingual-uncased-sentiment | **73.22** | **54.67** |
| Pooling ERM (MLP) | bert-base-multilingual-uncased-sentiment | 73.01 | 53.33 |
| DI-ERM (MLP) | bert-base-multilingual-uncased-sentiment | **73.18** | **55.07** |
| DI-ERM (Domain2Vec) | bert-base-multilingual-uncased-sentiment | 73.19 | 54.67 |

```
Domain "photo", text prompt:  "a photo"

Domain "art painting", text prompt:  "an oil painting"

Domain "cartoon", text prompt:  "a colorful cartoon"

Domain "sketch", text prompt:  "a pencil sketch"
```

Figure 4: Example of style-specific text prompts used as domain descriptions.

Notably, we observe that the performance gains from DI-ERM tend to diminish as model capacity increases. For the largest models (e.g., CLIP ViT-L/14 and DINOv2 ViT-L/14), the improvement is marginal or saturates. This trend is also observed by various empirical works, e.g. Cho et al. [2023].

Table 10: Domain generalization results on PACS using models from the CLIP and DINOv2 families. DI-ERM achieves improved accuracy over pooling ERM in most configurations, particularly for mid-sized models.

| Model | Algorithm | PAC $\rightarrow$ S | ACS $\rightarrow$ P | CSP $\rightarrow$ A | SPA $\rightarrow$ C | Test Avg Acc |
|---|---|---|---|---|---|---|
| CLIP: vitb32 | Pooling ERM (linear) | 86.97 | 99.58 | 95.90 | 97.48 | 94.98 |
| | DI-ERM (linear) | **88.06** | **99.64** | **96.29** | 97.48 | **95.37** |
| CLIP: vitb16 | Pooling ERM (linear) | 90.89 | 99.70 | 97.51 | 98.76 | 96.70 |
| | DI-ERM (linear) | **91.09** | 99.70 | **97.61** | 98.76 | **96.79** |
| CLIP: vitl14 | Pooling ERM (linear) | 95.42 | 99.94 | 99.22 | 99.79 | 98.59 |
| | DI-ERM (linear) | 95.32 | 99.94 | **99.32** | 99.79 | 98.59 |
| DINOv2: vits14 | Pooling ERM (linear) | 79.82 | 85.81 | 93.55 | 91.34 | 87.63 |
| | DI-ERM (linear) | **80.45** | **90.00** | **94.09** | **91.60** | **89.04** |
| DINOv2: vitb14 | Pooling ERM (linear) | 87.27 | 95.45 | 97.66 | 94.67 | 93.76 |
| | DI-ERM (linear) | **87.35** | **96.53** | **98.05** | 94.50 | **94.11** |
| DINOv2: vitl14 | Pooling ERM (linear) | 92.29 | 96.41 | 98.14 | 97.48 | 96.08 |
| | DI-ERM (linear) | **92.42** | **97.37** | 98.10 | 97.48 | **96.34** |

