# OpenReview forum: "Domain Generalization: A Tale of Two ERMs"
_NeurIPS.cc/2025/Workshop/Reliable_ML — NeurIPS 2025 - Reliable ML Workshop_

### Official Review · Reviewer_D7PU · 2025-09-19
**This paper proposes a Domain Generalization method by incorporating partial domain information into ERM classifiers and theoretically proves that this method will outperform Pooling ERM in the posterior shift regime with empirical evidence.**

**Rating:** 7
**Confidence:** 4

**Review:**

Summary: This paper proposes a Domain Generalization method by incorporating partial domain information into ERM classifiers and theoretically proves that this method will outperform Pooling ERM in the posterior shift regime with empirical evidence.

Overall, this is a well-written paper. This proposes a DG method that uses partial domain information in the ERM model. Partial information is useful in posterior shift domains where $P_{Y|X, D}$ is not fixed compared to covariate shift. The authors prove that the risk of this method will lie somewhere between the full information case and the no information case (Pool ERM). The authors provide enough empirical evidence to support their claim.

Strengths:

1) Well written

2) Theoretical proof for their claims

3) Enough empirical studies

4) Practically sensible conjecture

Weaknesses:

They use the partial information about the domain to augment feature vectors. Ultimately, it comes down to using more features compared to Pool ERM. I do not find much novelty in this method. Furthermore, the partial knowledge about domains is withheld from the baseline methods, so, naturally, they will perform worse. Say, someone says that they want to add the domain information to the feature $X$ and then run pool ERM, won't that achieve the same performance as the proposed method?

Suggestion:

It would be nice if authors could address the concern in the weaknesses section.